# Improving the Mechanical Properties of Orthodontic Occlusal Splints Using Nanoparticles: Silver and Zinc Oxide

**DOI:** 10.3390/biomedicines11071965

**Published:** 2023-07-12

**Authors:** Ioan Barbur, Horia Opris, Horatiu Alexandru Colosi, Mihaela Baciut, Daiana Opris, Stanca Cuc, Ioan Petean, Marioara Moldovan, Cristian Mihail Dinu, Grigore Baciut

**Affiliations:** 1Department of Maxillofacial Surgery and Implantology, Iuliu Hatieganu University of Medicine and Pharmacy, 400012 Cluj-Napoca, Romania; drbarur@gmail.com (I.B.); mbaciut@umfcluj.ro (M.B.); daiana.prodan@umfcluj.ro (D.O.); cristian.dinu@umfcluj.ro (C.M.D.); gbaciut@umfcluj.ro (G.B.); 2Department of Medical Education, Division of Medical Informatics and Biostatistics, Iuliu Hatieganu University of Medicine and Pharmacy, 400012 Cluj-Napoca, Romania; 3Department of Polymer Composites, Institute of Chemistry Raluca Ripan, Babes-Bolyai University, 30 Fantanele Str., 400294 Cluj-Napoca, Romania; stanca.boboia@ubbcluj.ro (S.C.); marioara.moldovan@ubbcluj.ro (M.M.); 4Faculty of Chemistry and Chemical Engineering, Babes-Bolyai University, 11 Arany Janos Street, 400028 Cluj-Napoca, Romania; ipetean@chem.ubbcluj.ro

**Keywords:** methylmethacrylate, graphene, occlusal splint, mechanical properties, silver, zinc oxide, microscopy, electron, scanning

## Abstract

The goal of the current study was to determine the mechanical proprieties of polymethylmethacrylate (PMMA) and the improved compound, the graphene-based PMMA, with Zn and Ag and to compare the results. Scanning electron microscopy analysis of the samples before and after the mechanical test was conducted. The compression behavior, flexural properties, tensile strength, and shape of the samples were all investigated and compared between the variants of PMMA. Commercially available polymethylmethacrylate was used (Orthocryl^®^—Dentaurum, Ispringen, Germany) with the salt and pepper technique according to the manufacturer’s instructions to produce 20 samples for each mechanical trial with standard cylinders (4 mm diameter × 8 mm length) for compression, parallelepipedal prisms for flexing (2 mm × 2 mm × 25 mm) and flat samples for traction. There was no statistical difference in the mechanical proprieties of the samples evaluated, although there were values that could suggest significance. The graphene-based PMMA demonstrated good mechanical proprieties, like the commercially available PMMA, and appears promising for future clinical use based on its multiple advantages.

## 1. Introduction

Even though some new alloplastic chemicals show creative promise, polymethylmethacrylate (PMMA) is versatile and widely used [1]. PMMA was first used for full denture bases. Its biocompatibility, dependability, relative ease of manipulation, and minimal toxicity have attracted many medical professionals. PMMA is used in bone cement, contact and intraocular lenses, bone screw fixation, skull filler, and osteoporosis vertebrae stabilization [1].

Prosthodontics uses various polymers. PMMA is used to fabricate artificial teeth, denture bases, dentures, obturators, orthodontic retainers, temporary or provisional crowns, and dental prostheses. It is also used for occlusal splints, printed or milled casts, treatment planning dyes, and research tooth implantation. Due to its low density, attractiveness, cost-effectiveness, ease of manipulation, and adjustable physical and mechanical properties, PMMA is a desirable biomaterial for dental applications. In recent years, multiple chemical changes and mechanical reinforcing techniques using various fibers, nanoparticles, and nanotubes have been described to further improve PMMA’s capabilities (thermal properties, water sorption, solubility, impact strength, flexural strength) [2].

Occlusal splints rarely cure bruxism or occlusal-incisal attrition. Even if nocturnal bruxism is never abolished, the sturdy, smooth polymethyl methacrylate occlusal splint has performed well in other duties [3]. Separating the patient’s teeth prevents overnight occlusal abrasion and most bruxing effects. A smooth, stiff splint works better than an athletic mouth guard, which can enhance bruxing and clenching [4].

Orthodontics uses muscle-reconditioning splints to reduce the occlusal irregularities and micro-displacements of the mandible caused by tooth shape and position changes. The biting plate should be 1.25 mm thermoformed, semi-rigid, and flat on the back. In a lab, centric models should be used to make the neuromuscular reconditioning splint. The thermo-formed splint’s thickness is usually just enough to flatten its surface. Thermoformed splints are better than plastic plates and are fashioned from plaster models, which always have bubbles and other problems. Temporomandibular joint (TMJ) splints should be worn day and night in extreme cases [5].

Such splints aid TMJ and jaw pain treatment, according to the literature. Full-coverage flat-plane stabilizing devices and anterior repositioning splints have been compared in studies. Occlusal splints have a canine rise and incisal guiding cover malocclusions. Functional occlusal splints are sometimes used to treat bruxism. Since bruxism is a parafunctional habit, it must be recognized and treated accordingly [6].

PMMA has poor abrasion resistance, surface hardness, polymerization shrinkage, and low strength [7]. Despite the development of polymers for denture bases, PMMA is still the best material for removable prostheses and occlusal splints [8].

The use of reinforcing chemicals to improve the mechanical performance of the material is one approach [9]. Other methods include modifying the chemical structure of PMMA by copolymerization [10] or by adding chemical compounds such as cross-linking agents [11]. Cross-linked polymers demonstrate critical properties not only for dentistry but also in industrial applications [12].

PMMA is a thermoplastic linear polymer. It has a low capacity for moisture and water absorption. High mechanical strength, high Young’s modulus, and minimal elongation at the break characterize PMMA. It is one of the most scratch-resistant thermoplastics and one of the toughest [13].

Due to their size-dependent optical characteristics [14], noble metal particles, such as silver and gold, are of tremendous relevance. In addition, it has been proven that the antibacterial activity of silver nanoparticles contained in a polymer matrix can effectively inhibit the development of *Salmonella* spp., *Staphylococcus* spp., and *E. coli* spp. [15].

Zinc oxide (ZnO, or “zincite”) is a wide-bandgap semiconductor that is colorless and has an optical bandgap in the UV range. This makes it a good UV absorber, which is also environmentally friendly. It has a refractive index, thermal conductivity, and electrical conductivity that is much higher than those of PMMA. The combination of these two materials is useful for many things, such as coatings that do not reflect light, films that protect against UV light, transparent barrier/protective layers, and materials that do not catch fire [16].

The current study focused on medical applications of PMMA in orthodontics and orthognathic surgery, particularly on its use of occlusal splints, paying close attention to its physical characteristics, benefits, drawbacks, and consequences, as well as comparing it to improved, enriched compounds.

To improve both the mechanical and composite properties of PMMA for orthodontic use in occlusal splints, graphene oxide-Ag, and graphene oxide-Zn nanoparticles were also added to the PMMA resin. The goal of the current study was to determine the mechanical proprieties of these materials compared to commercially available PMMA and to silver and zinc-enriched PMMA. The compression behavior, flexural properties, tensile strength, and shape of the samples are all investigated and compared.

The peculiarity of this study is that the synthesis phase of the new materials was conducted utilizing graphene, which is nowadays used exclusively for experimental purposes only. At the same time, the samples were processed exactly as if they were in the oral cavity, mimicking the exact environment that causes mechanical tensions.

## 2. Materials and Methods

### 2.1. Manufacturing of Experimental Samples

#### 2.1.1. Commercial PMMA Samples

Commercially available polymethylmethacrylate was used (Orthocryl^®^—Dentaurum, Ispringen, Germany) with the salt and pepper technique according to the manufacturer’s instructions to produce 20 samples for each mechanical trial, including standard cylinders (4 mm diameter × 8 mm length) for compression, parallelepipedal prisms for flexing (2 mm × 2 mm × 25 mm) and flat samples for traction.

#### 2.1.2. Synthesis of Graphene Oxide PMMA

The graphene–silver oxide nanoparticles (GO-Ag) employed in this study were created by combining graphene oxide (GO) with AgNO_3_ in a mass ratio of 2:1 (at the Laboratory of Polymeric Composites, UBB-ICCRR, Cluj-Napoca, Romania). The precipitate was frozen and then freeze-dried until it became powder.

Graphene oxide nanoparticle—zinc oxide (GO-ZnO) was created (at the Laboratory of Polymeric Composites, UBB-ICCRR, Cluj-Napoca, Romania) by dissolving ZnO powder (GO:ZnO = 1:20 (g/g)) in 750 cm^3^ of distilled water and then adding 15 mL of NaOH (pH = 9.5). The mixture was ultra-sounded at a maximum speed for 75 min before being magnetically stirred at 50 °C. The precipitate was frozen and then freeze-dried to produce a homogenous powder.

For the good embedding of the graphene powders in the polymer matrix, they were lyophilized (Lyophilizate—Alpha 1-4LDPLUS, Martin Christ Gefriertrocknungsanlagen GmbH, Osterode am Harz, Germany).

The enhanced PMMA experimental samples were obtained from transparent polymethylmethacrylate used in orthodontics (Orthocryl^®^—Dentaurum, Ispringen, Germany). The control sample contained only polymethylmethacrylate (M), while P1 and P2 contained PMMA + 0.2% GO-Ag (P1) and PMMA + 1% GO-ZnO (P2).

The powders were obtained by mixing 250 g of PMMA with 0.5 g GO-Ag/2g GO-ZnO and 145 mL of ethyl alcohol p.a. 98%, by continuous mixing on a magnetic stirrer, until the complete evaporation of the alcohol.

### 2.2. Evaluation of Mechanical Properties

The samples were manufactured in accordance with ISO 4049:2019 [17] for mechanical assessment. To reduce the development of fractures and faults inside the material, silicone molds were employed throughout the preparation process.

Mechanical properties testing was performed at a room temperature of 23 °C using a universal testing machine (LR5K Plus, Lloyd Instruments Ltd., Bognor Regis, UK) with a maximum permissible capacity of 5KN. The data were processed using Nexygen software (version 4.0) [18]. The samples (*n* = 20) were polymerized according to the manufacturer’s conditions at specific dimensions according to ISO 4049:2019 [17].

Three different types of mechanical tests were performed:

The compressive strength test measured the rigidity of the material along with the compressive strength curve and the breaking point. The test results were rendered by Nexygen software during the testing of each compression sample, using a speed of 0.5 mm/min and a force of 50 N/s. The samples subjected to this test had a cylindrical shape with dimensions 6 mm high and 4 mm in diameter.

The bending resistance test, measuring bending resistance and Young’s Modulus, was performed using the three-point test. Samples in the shape of rectangular prisms with a thickness of 2 mm, a height of 2 mm, and a length of 25 mm were used. The lowering speed of the arm was 1 mm/min with a force of 50 N/min.

The tensile strength was measured by applying an increasing axial force to a test piece, usually until it broke, and recording the corresponding variations in the length of the sample. The measurements were performed based on the ASTM-D638 reference standard [19] (with a load force of 5 N and a stress of 1000 N/mm^2^). The test pieces used for the traction test had a flat rectangular shape with a thickness of 2 mm, a width of 6 mm, and a length of 40 mm, presenting a calibrated portion and two gripping ends for the testing device (2 mm × 10 mm × 10 mm).

### 2.3. Scanning Electron Microscopy

An FEI Inspect microscope (SEM-Inspect S, FEI Company, Hillsboro, OR, USA) S model, functional in high-vacuum and low-vacuum, with an accelerating voltage between 200 V and 30 kV, was used to conduct the scanning electron microscopy on the fillers and the samples (discs). The microscope had a backscatter electron detector and a CCD-IR infrared inspection camera, with image processing up to 4096 × 3536 pixels. Typically, a 500–1000 times magnification was used to acquire the photos.

### 2.4. Statistical Analysis

The statistical description and analysis of the mechanical properties reflected by the measured data were performed using JASP Version 0.16.3 (JASP Team, University of Amsterdam, The Netherlands) [20].

The distribution of the measured data was evaluated using Q–Q plots and by applying Shapiro–Wilk tests. 

Descriptive statistics were computed and reported. Regardless of the data distribution, median values were considered the robust measures of central tendency, and interquartile ranges (IQR) were considered the robust measures of data spread. For normally distributed data (*p* > 0.05 after the Shapiro–Wilk test), mean values and standard deviations (SD) were also considered acceptable measures of the central tendency and data spread, respectively. The minimum and maximum measurements were also reported as a reflection of the data spread.

For normally distributed data, ANOVA with Welch homogeneity correction was performed to compare mechanical measurements between the three investigated groups (Ag-PMMA, Zn-PMMA, and Control). For data with marked asymmetry (*p* < 0.05 after the Shapiro–Wilk test), non-parametrical Kruskal–Wallis tests were used for the same purpose. 

Post hoc paired testing was performed using Dunn’s Post Hoc comparisons.

For all the comparisons, the level of statistical significance was chosen at α = 0.05.

Boxplots, raincloud plots, as well as plots for means and their 95% CI, were used to visually display and compare the mechanical properties of the investigated materials.

## 3. Results

### 3.1. Compression

In Table 1, the descriptive statistics of the compression test could be observed, with only the control of Young’s Modulus of Compression with a *p* < 0.05.

In Figure 1 the rain cloud plots of the compression test can be observed with Young’s Modulus of Compression (MPa) and the tensile strength (MPa).

In Table 2 the comparison for the compression test can be seen, with a *p* < 0.05 for the Ag-Zn comparison and Young’s Modulus of Compression.

### 3.2. Bending/Flexural

In Table 3 the descriptive statistics for the bending test can be observed with a significant *p* (*p* < 0.05) for Young’s Modulus of Bending (MPa) and stiffness (N/M) for all the tested samples.

In Figure 2 the rain clouds plots for the bending test can be observed with Young’s Modulus of Bending, Load at Break, Maximum Bending Stress at maximum Load and Stiffness for the PMMA (control) and for the PMMA with Ag and PMMA with Zn samples.

In Table 4, a comparison can be observed between the samples and the statistical test chosen. There was statistical significance between the samples as a whole with the Ag-Zn-Control for Young’s Modulus of Bending (MPa) and the Load at Break (N). When comparing the samples two at a time, there was a statistically significant difference between Ag-Zn and Ag-Control when assessing Young’s Modulus of Bending (MPa). Additionally, for the Load at break (N), there was also a statistically significant difference for the Ag-Zn comparison. For the Zn-Control comparison, there was a statistically significant difference when comparing the Maximum Bending stress at the maximum load (MPa) and Stiffness (N/m).

### 3.3. Traction

In Table 5, the descriptive statistics of the bending test—Young’s Modulus of Traction and Elongation at fracture of the materials PMMA (control), PMMA enriched with Ag, and PMMA enriched with Zn samples—can be observed. There was statistical significance for the control PMMA samples and for Young’s Modulus of Traction (MPa).

In Figure 3, the rain cloud plots of Young’s Modulus of Traction and Elongation at fracture for the PMMA (control), PMMA with Ag, and PMMA with Zn samples are assessed.

Table 6 presents a comparison between Young’s Modulus of Traction and Elongation at fracture between the investigated materials (PMMA enriched with Ag, PMMA enriched with Zn, and control PMMA). There was a significant statistical difference when comparing all three samples Ag-Zn-Control (*p* < 0.05). When comparing two samples at a time, there was a statistically significant difference for the Ag-Control and Zn-Control comparisons for Young’s Modulus of Traction (MPa).

### 3.4. SEM Analysis

The morphology of the samples was examined by Scanning Electron Microscopy (SEM) at low magnification for the observation of major morphologic features and a microstructure overview, at the average magnification for precise morphological features, and at high magnification for microstructural details.

The PMMA sample before the flexural strength test, as shown in Figure 4, presented a compact microstructure formed by PMMA particles’ diffusion to each other, which formed strong connection necks due to the photopolymerization process. However, several pores occurred with polyhedral shapes and sizes in the range of 80–200 µm as observed at a 500× magnification. A pore detail observed at 1000× magnification is shown in Figure 4; evidence shows a strong connection between photo-polymerized PMMA particles. Fine PMMA filaments were formed on the pores’ surface, proving the efficacy of the consolidation process. A high magnification detail (5000×) revealed the neck between two former PMMA particles in the middle-lower side of the image, evidencing the material diffusion from the left to the right. The left upper side of the image revealed a pore formation tendency. This complex structure might be very robust because of the effort dissipation through the walls of the pores that increased the material’s tenacity.

The fracture microstructure after the flexural strength test was also investigated by SEM microscopy, and the obtained results are presented in Figure 4. There occurred complex solicitations such as lateral compression on the upper side of the testing specimen and elongation on the lower side. The effort was dissipated through the pores and generated complex micro solicitations over the PMMA necks within the material. The failure started to appear on the lower side due to the progressive elongation fact sustained by the breaking line and observed at low magnification. An average magnification revealed PMMA necks failure in the proximity of the surface of the pores and their internal rupture. It occurred due to the necks’ elongation over the maximum resistance point as observed at high magnification (5000×); the middle lower side of the image demonstrated a broken neck with the fracture margin as a bent plastic sheet and as a consequence of intensive elongation effort.

On the upper side, the compression effort induced pore shrinkage and a material densification that resisted longer. The failure initiated the lower side of the specimen, which progressively propagated into the material until it reached the dense layer, which was suddenly exposed on the elongation effort and caused the complete breaking of the sample forming a distinct elongated margin as observed on the right side of the SEM image at low magnification, Figure 4. Thus, the overall behavior of the PMMA sample demonstrated a good tenacity but relatively low flexural strength. The situation might be improved by the addition of small micro-sized filler material.

PMMA reinforced with Graphene Oxide (GO) and an Ag ultrafine filler has an improved structure. The macroscopic aspect observed at a low magnification of 100× revealed a compact material with fine grains that were very well embedded into the polymeric matrix, Figure 5. The microstructure aspect, 500×, revealed a more compact structure with smaller pores (e.g., polyhedral shapes and sizes of about 50–100 μm) and an enhanced diffusion neck that assured better compaction. PMMA filaments were less visible, indicating better incorporation of the polymerized matrix onto the diffusion necks. This fact could be explained by the uniform distribution of ultrafine filler particles. GO acts as a microstructural moderator due to its relative lamellar shape, and it is well embedded onto the PMMA matrix giving more robustness, while silver submicron and nano (e.g., 80–650 nm in diameter) particles are uniformly distributed within the consolidated diffusion necks, giving more toughness, 1000× Figure 5.

GO and AG ultrafine filler increase the flexural strength of the PMMA composite, improving the fracture aspect, Figure 5. The macro-structural aspect revealed a smooth composite surface in the upper side of the image observed at 100× and a regular aspect of the fracture. At an average magnification of 500×, it showed a relatively undeformed microstructure (lower side of the image) which indicated the better dimensional constancy of the sample under deformation efforts. The lower side of the sample was less elongated, and the upper side was less compressed. This stability is gained due to the uniform distribution of silver nanoparticles and GO submicron sheets that prevented deformation under stress. Sample breaking began at the lower sample side due to the elongation effort, which caused the local delamination of the ultrafine filler particles and intergranular failure. It suddenly spread onto the vertical splitting sample in two parts connected by a 30 μm layer on the upper side which was significantly deformed before final failure.

GO + Zn ultrafine filler addition into PMMA influenced the composite microstructure. The macroscopic aspect of the sample, 100× Figure 6, presented large pores with polyhedral shapes and sizes between 100 and 300 μm randomly distributed onto a more compact area. The compact areas presented a heterogeneous microstructure, 500×, that presented incompletely polymerized PMMA spherical particles of about 30 μm embedded into a compact polymerized matrix with ultrafine filler particles uniformly distributed. At high magnification, 1000×, the expansion of polymerized PMMA particles could be observed as a polyhedral grain with diffusion contact on each side without neck formations. The cohesion between the diffused polymer grains could be enhanced by graphene oxide lamellas that assured proper microstructural cohesion. Zinc ultrafine particles presented rounded shapes and sizes ranging from 95 to 200 nm, which were distributed uniformly into the polyhedral grains.

The failure of the GO + Zn PMMA composite is a matter of, unfortunately, disposing of the large pores and unpolymerized PMMA spherical particles, which acted as micro-cracks and an initiator during the solicitation efforts, 100× Figure 6. Both the elongation of the lower sample size and compression of the upper sample side broke the weak cohesion within the pores and incompletely polymerized PMMA particles. The lower side of the sample failed first, and the upper side resisted a little longer due to strong cohesion sites situated on the right and left side of the incomplete polymerized particles, as observed in Figure 6 ×100. The microstructural aspects observed at 500× and 1000× revealed the intracrystalline failure of the compact grains due to the supplementary effort caused by the lack of sustenance from the pores and incompletely polymerized PMMA particles. In addition to these limitations, the microstructural aspects were improved by the GO + Zn addition that partly stabilized the microstructure preventing large deformations and increasing the cohesion between well diffused PMMA grains.

## 4. Discussion

Bacterial adhesion and plaque in the oral cavity are linked to several disorders, including stomatitis caused by dental prosthesis, which leads to fungal development, dental caries, and periodontal disease. It has always been extremely difficult to create and modify the physicomechanical characteristics of biomaterials and to induce antibacterial properties.

In a study where graphene was added to PMMA and the flexural and elastic modulus were measured, it was discovered that each sample enriched with graphene exhibited significantly increased flexural strength and elastic modulus values [21]. The SEM analysis of the samples revealed that the G-PMMA had an inhomogeneous fracture morphology [21].

In another paper comparing PMMA and CAD/CAM, the authors discovered that the flexural strength, flexural modulus, and impact strength were greater in the CAD/CAM group than in the heat-cured group [22]. There was also a review study with similar conclusions [23].

In recent years, nanodiamonds were added to the acrylic denture foundation to increase flexural strength and surface roughness at low concentrations but decreased impact strength [24].

PMMA has a high Young’s Modulus and a low elongation at breakage and does not shatter on rupture. It was considered one of the toughest thermoplastics with high scratch resistance [13].

When comparing PMMA with the nanocomposite PMMA enriched with boron nitride/silver, the compressive strength and flexural strength improved by over 50% [25].

Other studies using graphene silver nanoparticles to improve PMMA’s mechanical proprieties have shown that in a concentration of 1wt%, the nanoparticles have a beneficial effect on the resin’s modulus of resilience but decrease the ductility. On the other hand, a 2wt% concentration improved resilience and reduced ductility [26].

Other studies have compared the mechanical proprieties of PMMA when enriched with silver nanoparticles and PMMA and have demonstrated a decrease in the bending strength significantly (*p* < 0.05) while still conforming with the ISO standard for denture base materials [27].

A study evaluating PMMA zinc–oxide composite proprieties has shown that increasing the ZnO content increases the impact strength of the blend [28]. Another study demonstrated that adding ZnO to PMMA increased compressive strength [29]. Other papers have confirmed these findings with similar results when investigating mechanical properties, including the impact, flexural, and hardness of reinforced PMMA ZnO nanostructures [30,31].

In a paper comparing the mechanical proprieties of PMMA enriched with a HA nanocomposite, using wear, compression, and three-point bending were applied. It was found that the wear rate decreased in the enriched nanocomposite. The compression test presented an added 2.5% in compressive strength, yield strength, and modulus. It also decreased elongation at break [32].

Combining polycarbonate and PMMA (50/50) by the injection molding process has been shown that the melt temperature can significantly impact the resulting composite’s tensile and impact strength [33].

PMMA enriched with ɣ-methacryloxypropyltrimethoxysilane determined a material with slightly worse flexural strength and flexural modulus [34].

PMMA with silica has been shown to have decreased tensile strength and elongation at break when prepared by solution blending, while solution polymerization has shown to increase mechanical performance when compared to PMMA [35].

Additionally, newer studies have enriched PMMA with Zirconium Oxide increases in flexural strength, although these findings need to be verified further [36].

The mechanical results of this paper confirm findings in the literature that there is some significant difference between the samples tested (PMMA–PMMA with graphene oxide and silver–PMMA with graphene oxide and Zn) when comparing the mechanical results. The significance could be observed for the modulus of compression between Ag and Zn. Regarding the bending tests, Young’s Modulus had a significant difference for all three groups when compared and for the Ag-Zn and Ag-PMMA comparisons. When Traction was evaluated, significance was found for all three samples when compared and for the pairs: Ag–PMMA and Zn–PMMA.

As an alternative to PMMA, 3D printing could be an option, with excellent aesthetics but lower mechanical properties compared to other materials, and milling using CAD-CAM devices could provide better results long term, aesthetically and mechanically [37]. Depending on the specific commercial model, 3D printing can achieve different types of resulting materials, which might be stronger or not, when compared to conventional PMMA [38].

More research is needed to better understand the complete properties of the materials. Additionally, there is a real need to develop a standard technique for mixing polymethylmethacrylate that is not dependent on the operator.

## 5. Conclusions

PMMA is a versatile material with extensive use in the medical field, especially dentistry. It can be modified, and its properties can be altered to have better mechanical properties using Zn and Ag.

PMMA materials, when modified with graphene–silver oxide nanoparticles (GO-Ag) and graphene–zinc oxide nanoparticles (GO-ZnO), in our study, demonstrated excellent mechanical proprieties, like the commercially available PMMA, and appear promising for future clinical use based on its multiple advantages.

PMMA materials still hold a very important role in orthodontic treatment due to their versatility, cost, and ease of manipulation, although some of their proprieties could be enhanced and modified to satisfy current clinical needs.

## Figures and Tables

**Figure 1 biomedicines-11-01965-f001:**
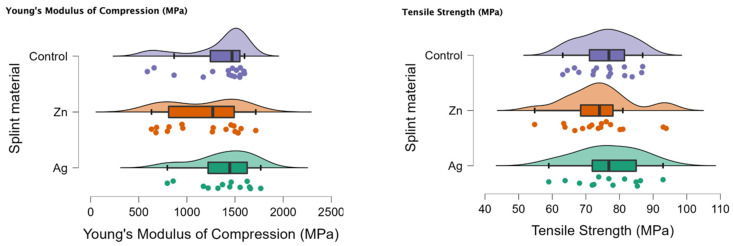
Rain cloud plots of Young’s Modulus of Compression and tensile strength for the PMMA (control), PMMA with Ag and PMMA with Zn.

**Figure 2 biomedicines-11-01965-f002:**
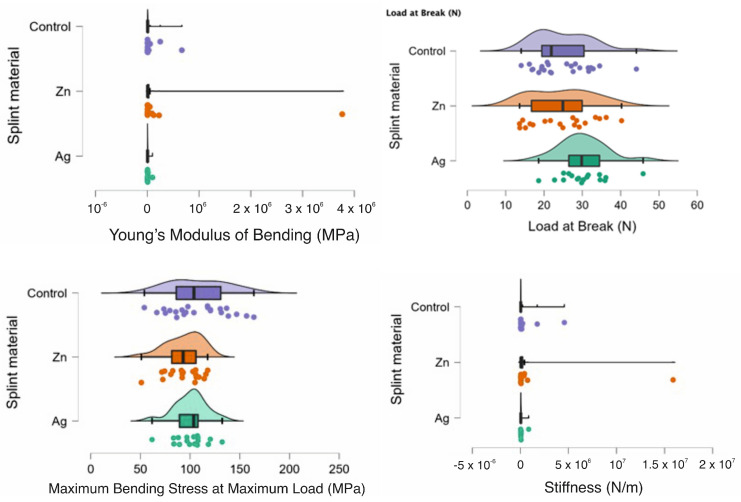
Rain cloud plots of Young’s Modulus of Bending, Load at Break, Maximum Bending Stress at maximum Load and Stiffness for the PMMA (control) and for PMMA with Ag and PMMA with Zn.

**Figure 3 biomedicines-11-01965-f003:**
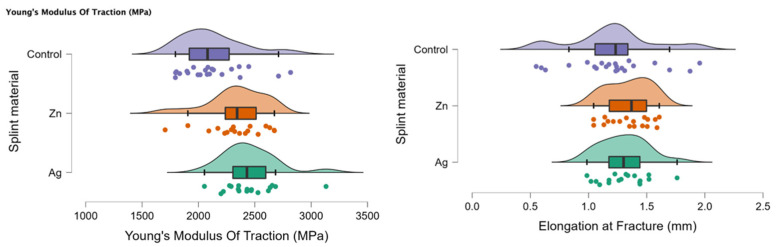
Rain cloud plots of Young’s Modulus of Traction and Elongation at fracture for the PMMA (control), PMMA with Ag and PMMA with Zn.

**Figure 4 biomedicines-11-01965-f004:**
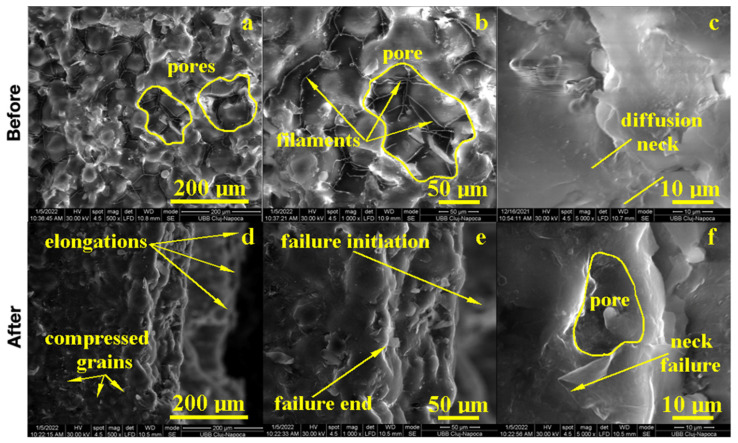
SEM imaging of the PMMA (control) sample before: (**a**) 500×, (**b**) 1000×, (**c**) 5000× and after flexural strength tests: (**d**) 500×, (**e**) 1000× and (**f**) 5000×.

**Figure 5 biomedicines-11-01965-f005:**
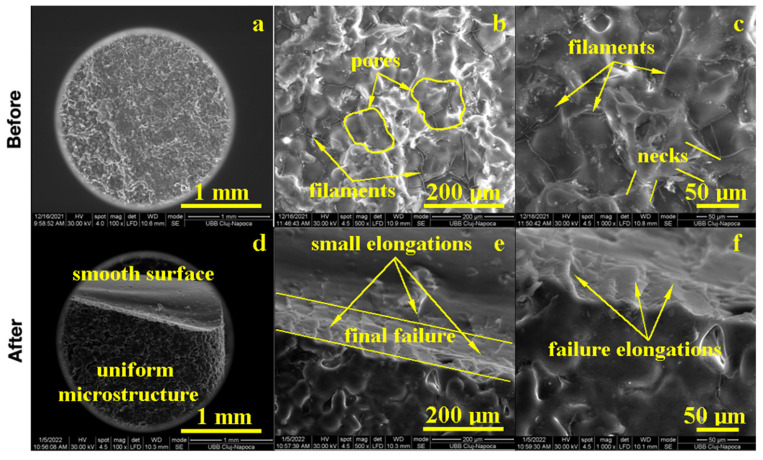
SEM imaging of the PMMA enriched with Ag before: (**a**) 100×, (**b**) 500×, (**c**) 1000× and after flexural strength tests: (**d**) 100×, (**e**) 500× and (**f**) 1000×.

**Figure 6 biomedicines-11-01965-f006:**
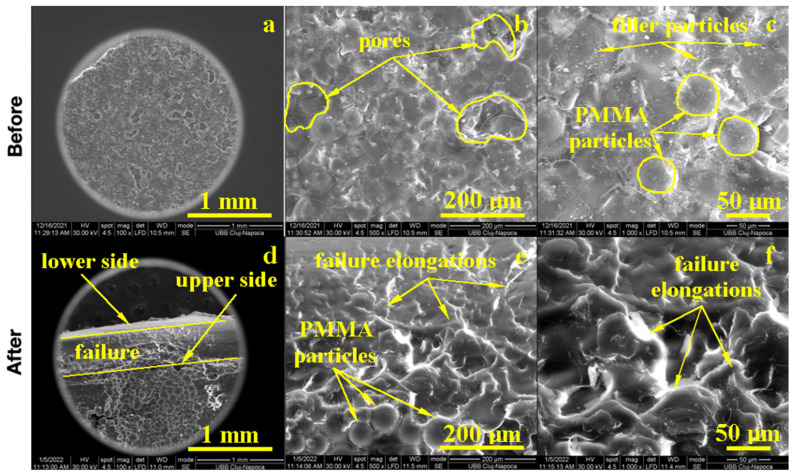
EM imaging of the PMMA enriched with Zn before (**a**) 100×, (**b**) 500×, (**c**) 1000× and after flexural strength tests: (**d**) 100×, (**e**) 500× and (**f**) 1000×.

**Table 1 biomedicines-11-01965-t001:** Descriptive statistics of the compression test—Young’s Modulus of Compression and Tensile Strength of the materials PMMA (control), PMMA enriched with Ag, PMMA enriched with Zn. (SD—standard deviation, IQR—interquartile range).

	Young’s Modulus of Compression (MPa)			Tensile Strength (MPa)		
	Ag	Zn	Control	Ag	Zn	Control
Valid N	13	16	16	13	16	16
Shapiro–Wilk *p*-value	0.271	0.069	<0.001	0.992	0.606	0.586
Median	1445.47	1269.19	1468.61	76.84	74.05	76.85
Mean	1377.07	1169.98	1326.18	76.53	74.05	75.51
SD	301.12	371.98	331.97	9.69	10.12	7.49
IQR	405.81	680.58	307.19	13.04	9.61	10.40
Minimum	797.06	632.75	589.36	58.97	54.77	63.12
Maximum	1767.32	1715.57	1598.58	92.95	93.87	86.89

**Table 2 biomedicines-11-01965-t002:** Comparison of Young’s Modulus of Compression and tensile strength between the investigated materials (PMMA enriched with Ag, PMMA enriched with Zn, and control PMMA).

	Young’s Modulus of Compression (MPa)	Tensile Strength (MPa)
	ANOVA *p*-values	
Ag-Zn-Control	0.259	0.799
	Dunn Post Hoc *p*-values	
Ag-Zn	0.075	0.190
Ag-Control	0.374	0.394
Zn-Control	0.118	0.259

**Table 3 biomedicines-11-01965-t003:** Descriptive statistics of the bending test—Young’s Modulus of Bending, Load at Break, Maximum Bending Stress at maximum Load and Stiffness of the materials PMMA (control), PMMA enriched with Ag, PMMA enriched with Zn. (SD—standard deviation, IQR—interquartile range).

	Young’s Modulus of Bending (MPa)			Load at Break (N)			Maximum Bending Stress at Maximum Load (MPa)			Stiffness (N/m)		
	Ag	Zn	Control	Ag	Zn	Control	Ag	Zn	Control	Ag	Zn	Control
Valid N	17	19	23	17	19	23	17	19	23	17	19	23
Shapiro–Wilk *p*-value	<0.001	<0.001	<0.001	0.699	0.321	0.174	0.800	0.220	0.833	<0.001	<0.001	<0.001
Median	5570.76	6842.74	7092.49	29.82	24.92	21.91	103.63	92.96	103.81	45,978.47	60,589.80	41,222.01
Mean	11,908.41	228,844.87	50,237.74	30.169	24.72	24.88	100.53	93.82	108.04	101,873.52	977,346.56	330,098.91
SD	23,088.49	858,455.75	144,037.90	6.24	8.28	7.40	16.47	17.96	29.96	197,193.30	3,616,000	991,464.40
IQR	3572.37	23,655.77	7143.04	8.02	13.24	11.01	18.63	24.29	44.77	32,255.68	192,847.16	25,595.48
Minimum	3599.78	4573.92	4115.90	18.60	13.60	14.04	61.71	50.97	53.94	30,268.68	34,481.91	25,276.75
Maximum	100,798.61	3,766,000.00	668,077.42	45.86	40.24	44.10	132.29	117.67	164.11	864,222.07	15,890,000	4,575,000

**Table 4 biomedicines-11-01965-t004:** Comparison of Young’s Modulus of Bending, Load at Break, Maximum Bending Stress at maximum Load and Stiffness alongside Dunn’s Post Hoc comparisons for PMMA (control), PMMA with Zn and PMMA with Ag.

	Young’s Modulus of Bending (MPa)	Load at Break (N)	Maximum Bending Stress at Maximum Load (MPa)	Stiffness (N/m)
Applied Test	Kruskal–Wallis	ANOVA	ANOVA	Kruskal–Wallis
	*p*-values			
Ag-Zn-Control	0.071	0.032	0.167	0.300
	Dunn Post Hoc *p*-values			
Ag-Zn	0.029	0.019	0.165	0.287
Ag-Control	0.017	0.014	0.286	0.185
Zn-Control	0.443	0.496	0.051	0.063

**Table 5 biomedicines-11-01965-t005:** Descriptive statistics of the bending test—Young’s Modulus of Traction and Elongation at fracture of the materials PMMA (control), PMMA enriched with Ag, PMMA enriched with Zn. (SD—standard deviation, IQR—interquartile range).

	Young’s Modulus of Traction (MPa)			Elongation at Fracture (mm)		
	Ag	Zn	Control	Ag	Zn	Control
Valid N	18	18	22	18	18	22
Shapiro–Wilk *p*-value	0.169	0.285	0.046	0.831	0.268	0.397
Median	2430.55	2343.13	2081.06	1.30	1.37	1.23
Mean	2453.66	2337.33	2126.85	1.30	1.34	1.21
SD	240.97	259.39	279.89	0.202	0.19	0.37
IQR	291.41	273.43	353.65	0.27	0.32	0.28
Minimum	2053.22	1704.73	1795.76	0.99	1.04	0.55
Maximum	3131.94	2673.89	2817.36	1.76	1.61	1.96

**Table 6 biomedicines-11-01965-t006:** Comparison of Young’s Modulus of Traction and Elongation at fracture between the investigated materials (PMMA enriched with Ag, PMMA enriched with Zn, and control PMMA).

	Young’s Modulus of Traction (MPa)	Elongation at Fracture (mm)
	ANOVA *p*-values	
Ag-Zn-Control	0.001	0.324
	Dunn Post Hoc *p*-values	
Ag-Zn	0.174	0.274
Ag-Control	<0.001	0.150
Zn-Control	0.007	0.048

## Data Availability

The data presented in this study are available on request from the corresponding author. The data are not publicly available due to privacy concerns.

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
