# Peer review of "Improving the Mechanical Properties of Orthodontic Occlusal Splints Using Nanoparticles: Silver and Zinc Oxide"

_biomedicines, 2023, doi:10.3390/biomedicines11071965_

Round 1

Reviewer 1 Report

Here, Barbur and colleagues report a study on PMMA mechanical characteristics doped by mixtures of either graphene oxide/Ag or graphene oxide/Zn intended for orthodontic occlusal procedures.

After careful consideration, I regret to inform that, in my opinion, the manuscript is not appropriate for publication because of the following reasons:

  • The article topic is not innovative, several scientific papers already described the proprieties of composite materials containing the structural elements reported in the manuscript; some of these are also mentioned in the reference section by the Authors.
  • There is no evidence about the improvement of the composite mechanical characteristics in comparison with commercially available products as stated by the Authors in the conclusion section too.
  • Some paper sections are not clear enough and an extensive article review is strongly suggested to proceed with a new submission.

More in details:

  • The introduction chapter does not provide a sufficient background about the state of the art of modified PMMA studies regarding the mechanical characteristic details useful in orthodontic occlusal strategies. This part sounds like a general description about the medical application of PMMA with a few remarks on the actual paper subject.
  • Line 115: more details regarding the polymer (i.e., molecular weight) and a brief description of the “salt and pepper” technique used in the study should be provided.
  • Line 123-131: more details about the procedure to obtain the composites should be described (i.e., GO and general material vendors; freeze-drying instrument description; was the GO-Ag nanoparticles obtained by a pre-mixed suspension? Have the nano-particles been characterized?)
  • Line 132: what are the PMMA characteristics (i.e. molecular weight)? Why did the Authors use a PMMA obtained from another supplier?
  • Table 1: is the control sample the commercially available PMMA or the polymer that the Authors used for producing the mixtures? This point results not clear in the whole results section. Furthermore, the abbreviation meanings should be described.
  • Figure 5: Although the statistical analyses revealed a certain degree of significance in the experiment, may the Authors explain why the points are so scattered?
  • SEM analyses: it’s hard to correlate the result descriptions with the pictures themselves, the Authors should use arrows or squares allowing a better understanding of their statements. Would have been possible a magnification increment to show the nanostructures (i.e. nano-particles).
  • Some typos to be corrected.

To be improved

Author Response

Response to reviewer 1

Here, Barbur and colleagues report a study on PMMA mechanical characteristics doped by mixtures of either graphene oxide/Ag or graphene oxide/Zn intended for orthodontic occlusal procedures.

After careful consideration, I regret to inform that, in my opinion, the manuscript is not appropriate for publication because of the following reasons:

Point 1

The article topic is not innovative, several scientific papers already described the proprieties of composite materials containing the structural elements reported in the manuscript; some of these are also mentioned in the reference section by the Authors.

There is no evidence about the improvement of the composite mechanical characteristics in comparison with commercially available products as stated by the Authors in the conclusion section too.

Some paper sections are not clear enough and an extensive article review is strongly suggested to proceed with a new submission.

Response 1:

We would like to thank you for your consideration. We believe to the best of our knowledge and research analysis that there is no current paper that evaluates the PMMA enriched with graphene (which is an experimental material only) for proposes such as any clinical use (e.g. orthodontic occlusal splint). On our extensive investigation prior to engaging into this inquiry we have not found any graphene enriched PMMA which was tested in a similar way for stress like the oral cavity. There are of course papers which hint towards the subject, but this specific angle has not yet been researched.

To our knowledge the publication of negative results is not discouraged by the scientific community and in fact is encoloured because one could not have a proper view of a subject of not all research is published. We strongly believe that there is a high risk of bias into considering not publishing research which does not have statistical significance.

The strength of this study, we strongly believe, that is the research quality, the reproducibility and the entirely new concept.

Point 2

More in details:

The introduction chapter does not provide a sufficient background about the state of the art of modified PMMA studies regarding the mechanical characteristic details useful in orthodontic occlusal strategies. This part sounds like a general description about the medical application of PMMA with a few remarks on the actual paper subject.

Response 2:

There is little to no research into the specifics of the materials used for orthodontic splints and we reinstate that this area of research is in need of more research and analysis.

Point 3:

Line 115: more details regarding the polymer (i.e., molecular weight) and a brief description of the “salt and pepper” technique used in the study should be provided.

Response 3:

The “salt and pepper” technique is something that is widely known in the dentist, orthodontist and dental technician world and is even described in the prospect materials and instructions for use of the material. We did not include them since the information we found that it adds up little to the research paper. For your consideration we will add the information below and if you recommend to add it in the paper we will gladly will:

“Salt & Pepper” or “Sprinkle” Technique:

Apply first a layer of powder (polymer), then a layer of fluid (monomer) and continue applying alternate layers of polymer and monomer in slight pendulum movements. Only apply as much fluid as the powder can soak up. The material should not flow away. To avoid the formation of air pockets under the screws, apply a slightly larger amount of fluid to begin with so that the acrylic spreads under the screws. Finish application with a layer of powder. Apply enough powder so that the top layer is dry. This ensures a precise fit and minimal shrink.

Point 4:

Line 123-131: more details about the procedure to obtain the composites should be described (i.e., GO and general material vendors; freeze-drying instrument description; was the GO-Ag nanoparticles obtained by a pre-mixed suspension? Have the nano-particles been characterized?)

Response 4:

Thank you so much for your feedback. The material and methods chapter has been reorganized to better reflect the matters described.

Point 5:

Line 132: what are the PMMA characteristics (i.e. molecular weight)? Why did the Authors use a PMMA obtained from another supplier?

Response 5:

We would like to thank you for your attentive review process. We checked with the research team and indeed there is an error in writing, all the materials used for mixing were procured from Orthocryl® - Dentaurum, Ispringen, Germany. We greatly appreciate your input.

Point 6:

Table 1: is the control sample the commercially available PMMA or the polymer that the Authors used for producing the mixtures? This point results not clear in the whole results section. Furthermore, the abbreviation meanings should be described.

Response 6:

Thank you very much for pointing these things out. We only used Orthocryl® - Dentaurum, Ispringen, Germanyfor both the control samples and for the experimental materials. We added the meaning of the abbreviations.

Point 7:

Figure 5: Although the statistical analyses revealed a certain degree of significance in the experiment, may the Authors explain why the points are so scattered?

Response 7:

Thank you so much for your remark. We strongly consider that the lack of homogeneity of the materials (especially the ones which are hand mixed with graphene) is to the lack of a standardized procedure or even a mechanical component of the mixing of the materials. Salt and pepper technique is something that is very operator dependent. We believe that the lack of this homogeneity has produced some samples which are outliers when it comes to the evaluation of the mechanical proprieties. Transparency and lack of bias is very important for us and that is why we did not alter the results by excluding these samples. We strongly believe that this adds more to the community.

Point 8:

SEM analyses: it’s hard to correlate the result descriptions with the pictures themselves, the Authors should use arrows or squares allowing a better understanding of their statements. Would have been possible a magnification increment to show the nanostructures (i.e. nano-particles).

Response 8:

Thank you very much for your feedback. All the SEM figures have been reorganized to better reflect and understand what the authors tried to present. We have also added highlights on the figures.

Point 9:

Some typos to be corrected.

Response 9:

Thank you very much for your consideration. We have thoroughly revised the article for typos. Thank you very much.

Reviewer 2 Report

The paper focuses on orthodontic materials based on  polymethyl- methacrylate that are reinforced with graphene oxide-silica and graphene oxide-zinc nanoparticles. The paper falls with the scope of the journal.

My comments/suggestions are shown blow:

The novelty of the work should be further articulated.

A number of figures are not essential and should be removed from the main body of the manuscript 1,2,3,4,6, 8.

Statistical analysis on mechanical data of the data should be provided.

Figure 7 is hard to discern.

The authors should provide information regarding the toxicity of their materials.

The conclusion section is not informative and should be thoroughly revised.

Author Response

The paper focuses on orthodontic materials based on  polymethyl- methacrylate that are reinforced with graphene oxide-silica and graphene oxide-zinc nanoparticles. The paper falls with the scope of the journal.

My comments/suggestions are shown blow:

Point 1:

The novelty of the work should be further articulated.

Response 1:

Thank you so much for your review. The novelty of the research is that we use an experimental material (graphene) for the development of a new material to be tested for further use in orthodontics as a splint material.

Point 2:

A number of figures are not essential and should be removed from the main body of the manuscript 1,2,3,4,6, 8.

Response 2:

Thank you very much for your critical appraisal of our paper. We have removed the unnecessary figures that you suggested, and we agree that they add little to the text.

Point 3:

Statistical analysis on mechanical data of the data should be provided.

Response 3:

Thank you very much for your review. All the mechanical data provided have been analyzed thoroughly. We have not included the raw data due to the fact that it would overflow the article and it would be rather cumbersome for the reader to find the information.

Point 4:

Figure 7 is hard to discern.

Response 4:

Thank you very much for your input. We have modified the figure accordingly to be more readable and accessible to the reader.

Point 5:

The authors should provide information regarding the toxicity of their materials.

Response 5:

Thank you very much for your review. The toxicity of the commercial material is provided by the manufacturer, and we believe that it would not add much for the scientific purpose of this research paper. We are currently studying the toxicity of these materials and we have in plan to publish this information as soon as we get the results. Thank you very much that you anticipated the next phase of our research, and we hope you are eager to find the conclusions of it.

Point 6:

The conclusion section is not informative and should be thoroughly revised.

Response 6:

Thank you very much for your feedback. We have revised the conclusion section thoroughly to emphasize the results of the paper.

Reviewer 3 Report

This study compared the mechanical proprieties of polymethylmethacrylate (PMMA) and the improved compound (PMMA modified with graphene-silver oxide nanoparticles and  graphene-zinc oxide nanoparticles) through a scanning electron microscopy analysis. The manuscript contains seven keywords, twelve figures, six tables, and forty references. Overall, it is a correct, complete, and well-conducted paper.

General comments
The main strength of this study is the synthesis of new materials applying graphene, a material which is nowadays used exclusively for experimental purposes only.
The study is methodologically well-designed. The equipment used and the techniques developed are duly described, making the procedures performed reproducible. The statistical analysis is appropriate according to the approach of the study. The results are well presented, being easy to read and interpret them. In the discussion, the results of this study are adequately contrasted with those obtained by other researchers. A good justifying explanation of the results is also provided.

Some remarks are made on different sections of the manuscript.

Keywords
The manuscript presents seven keywords. For keywords, where possible, please use Medical Subject Headings terms (MeSH Terms). Avoid the use of abbreviations as keywords. Strictly, only “occlusal splint”, “silver”, and “zinc oxide” are MeSH terms. Some alternative MesH terms proposed could be “methylmethacrylate” better than “polymethylmethacrylate”, and “microscopy, electron, scanning” rather than “scanning electron microscopy”. Nevertheless, these suggestions about keywords are optional, not mandatory.

Other manuscript sections
Line 68. Abbreviations and acronyms, even well-known, should be explained the first time they are used, eg. “TMJ”.
Line 98. The names of the microorganisms must be italicized. Do not just give the genus name, add the word "species" after it.
Figures 1, 2, and 3 are not cited in the text. Please, include them.

References
Total number of the manuscript references: 40.
The reference format does not exactly match the journal’s reference format (ACS style guide). The references should be checked carefully to transcribe them accurately. In some references,  the full journal name is used; in others, the abbreviated journal name. According to the journal’s guidelines, for journal articles, use the abbreviated journal name not the full journal name. Please, correct this.
For further information about the reference format proposed by the journal, please, consult the following link: https://www.mdpi.com/journal/biomedicines/instructions

Figures
Total number of the manuscript figures: 12.
The figures have appropriate figure legends. Nevertheless, please consider explaining the abbreviations in figure legends.  

Tables
Total number of the manuscript tables: 6.
The tables have appropriate titles and information. Please, consider adding a footer to explain the abbreviations where required. Please, use points instead of commas as decimal separators.

Author Response

Response to reviewer 3

Point 1:

This study compared the mechanical proprieties of polymethylmethacrylate (PMMA) and the improved compound (PMMA modified with graphene-silver oxide nanoparticles and  graphene-zinc oxide nanoparticles) through a scanning electron microscopy analysis. The manuscript contains seven keywords, twelve figures, six tables, and forty references. Overall, it is a correct, complete, and well-conducted paper.

Response 1:

Thank you very much for you feedback. We gladly appreciate your kind effort.

Point 2:

General comments

The main strength of this study is the synthesis of new materials applying graphene, a material which is nowadays used exclusively for experimental purposes only.

The study is methodologically well-designed. The equipment used and the techniques developed are duly described, making the procedures performed reproducible. The statistical analysis is appropriate according to the approach of the study. The results are well presented, being easy to read and interpret them. In the discussion, the results of this study are adequately contrasted with those obtained by other researchers. A good justifying explanation of the results is also provided.

Response 2:

We would like to appreciate your kind words. Thank you.

Point 3:

Some remarks are made on different sections of the manuscript.

Keywords

The manuscript presents seven keywords. For keywords, where possible, please use Medical Subject Headings terms (MeSH Terms). Avoid the use of abbreviations as keywords. Strictly, only “occlusal splint”, “silver”, and “zinc oxide” are MeSH terms. Some alternative MesH terms proposed could be “methylmethacrylate” better than “polymethylmethacrylate”, and “microscopy, electron, scanning” rather than “scanning electron microscopy”. Nevertheless, these suggestions about keywords are optional, not mandatory.

Response 3:

Thank you so much for the suggestion. We believe that it is important to use the right Mesh terms to properly value the research. We have modified the as suggested.

Point 4:

Other manuscript sections

Line 68. Abbreviations and acronyms, even well-known, should be explained the first time they are used, eg. “TMJ”.

Response 4:

Thank you very much for the feedback. We have completely revised the manuscript to properly mention the explanation of the abbreviation used.

Point 5:

Line 98. The names of the microorganisms must be italicized. Do not just give the genus name, add the word "species" after it.

Response 5:

Thank you very much. We have revised to text to be appropriately.

Point 6:

Figures 1, 2, and 3 are not cited in the text. Please, include them.

Response 6:

Thank you very much. We have decided with the authors and with the suggestions from other reviewers we removed them due to the fact that they do not add much value to the potential reader of this article.

Point 7:

References

Total number of the manuscript references: 40.

The reference format does not exactly match the journal’s reference format (ACS style guide). The references should be checked carefully to transcribe them accurately. In some references,  the full journal name is used; in others, the abbreviated journal name. According to the journal’s guidelines, for journal articles, use the abbreviated journal name not the full journal name. Please, correct this.

For further information about the reference format proposed by the journal, please, consult the following link: https://www.mdpi.com/journal/biomedicines/instructions

Response 7:

Thank you very much for your feedback. We have revisited the reference chapter to modify it and to be aligned with the MDPI and ACS style guidelines.

Point 8:

Figures

Total number of the manuscript figures: 12.

The figures have appropriate figure legends. Nevertheless, please consider explaining the abbreviations in figure legends. 

Response 8:

Thank you very much for your remarks. We have extensively reviewed the figures; we have removed some deemed unnecessary and have added appropriate abbreviations.

Point 9:

Tables

Total number of the manuscript tables: 6.

The tables have appropriate titles and information. Please, consider adding a footer to explain the abbreviations where required. Please, use points instead of commas as decimal separators.

Response 9:

Thank you very much for the feedback. We have revisited the tables and formatted them as suggested. Also, we added the necessary abbreviations.

Round 2

Reviewer 2 Report

Small improvements can be seen in the revised manuscript, however most of the points raised by the reviewers have not been addressed to the required depth.